# Learning Coarse-Grained Representations: An Exploration of Mutual Information via Hyperspherical Density

## Abstract

We revisit InfoMax for representation learning, using hyperspherical geometry with a non-parametric von Mises-Fisher kernel density estimator and differential entropy. This method is minimal with no asymmetry and trains stably. Results are competitive on smaller datasets such as CIFAR-10, STL-10 and LC25000, but lags behind modern baselines on ImageNet-1000. Experiments show that weakening the global entropy term consistently helps classification accuracy, suggesting that strict mutual information classification favors coarse grouping over fine discrimination.

## 1 Introduction

Self-supervised methods keep improving, but much of the progress has come from heuristics: architectural asymmetries, stop-grads, whitening, variance penalties, and carefully tuned augmentations. These choices work, yet the connection to first principles is often indirect. In this work, we revisit the InfoMax principle (Linsker, 1988b), and study a different estimator: a hyperspherical formulation with a non-parametric density and differential entropy. Our goal is to test how this theoretical backed formulation behaves, study what it is actually learning and where it breaks compared to modern SSL methods.

In this paper, we study an implementation of the InfoMax principle, while keeping it close to the mathematical foundation. We evaluate it with multiple datasets, showing that while it works well in smaller datasets, the learned representations end up representing coarser structures as the number of classes increases. We show that the low accuracy in ImageNet-1000 is the result of these coarser structures, not a fundamental failure in the training process. Additionally, we performed several experiments to determine how certain hyperparameters affect the classification accuracy, and found that weakening the global separation makes accuracy improvement slower, but leads to higher accuracy given enough epochs, while a stronger global separation leads to earlier higher accuracy but eventual learning stagnation.

## 2 Related Work

The InfoMax principle goes back to early neural coding work, where maximizing output entropy under simple constraints was proposed as a driver for useful features (Linsker, 1988a). It later grounded classic ICA via entropy maximization at the output layer (Bell & Sejnowski, 1995). A complementary line formalizes representation learning through explicit mutual information terms and variational bounds (Barber & Agakov, 2003; Belghazi et al., 2018).

Several deep methods are framed around maximizing mutual information between inputs and learned representations. Deep InfoMax pursues both global and local agreements (Hjelm et al., 2019). AMDIM extends this with multi-scale and patch-level objectives (Bachman et al., 2019). Contrastive predictive coding casts InfoMax through a classification view and the InfoNCE bound (van den Oord et al., 2019), a perspective that underlies later contrastive systems (Chen et al., 2020; He et al., 2020). A different but related path maximizes MI between cluster assignments for segmentation/cluster discovery (Ji et al., 2019).

Beyond InfoNCE, there are broader MI estimators and f-divergence bounds used in practice (Barber & Agakov, 2003; Belghazi et al., 2018; Nguyen et al., 2007). Work in information-theoretic learning also explores entropy and divergence criteria directly (Principe, 2010). Our approach sits in this space: we use a minimal, simple estimator on the hypersphere with differential entropy, keeping the objective close to first principles.

A parallel theoretical foundation argues for shaping information rather than maximizing it: preserve what predicts the task while discarding nuisance variability (Tishby et al., 2000). Many recent SSL objectives can be read as mixing an InfoMax drive (agreement across views) with implicit bottlenecks or regularizers—whitening, variance penalties, stop-grad, or architectural asymmetries (Zbontar et al., 2021; Bardes et al., 2021; Grill et al., 2020). Our study focuses on the pure Info-Max side and documents where that alone succeeds (coarse structure) and where extra bias appears necessary (fine-grained recognition).

## 3 APPROACH

Our method is based on the InfoMax principle. The objective of InfoMax is to maximize the mutual information between the source and the learned representations. The mutual information is defined as $I(X;Y) = H(Y) - H(Y|X)$. Calculating entropy in high-dimensional spaces is usually considered intractable. For that reason, we opted to restrict the representation geometry to a hypersphere. Under that geometry, we can use the von Mises-Fisher (vMF) kernel to estimate the density in a non-parametric way.

For unit vectors $\mathbf{x}_i, \mathbf{x}_j \in \mathbb{S}^{D-1}$ with cosine similarity $s_{ij} = \mathbf{x}_i^\top \mathbf{x}_j$, the von Mises–Fisher (vMF) kernel with concentration $\kappa$ and normalization constant $C_D(\kappa)$ is:

$$K_\kappa(\mathbf{x}_i, \mathbf{x}_j) = C_D(\kappa) \exp(\kappa\, s_{ij}), \tag{1}$$

Then, we estimate the global differential entropy as:

$$H_{\text{global}} = -\frac{1}{N} \sum_{i=1}^{N} \log \left( \frac{1}{N-1} \sum_{\substack{j=1 \\ j \neq i}}^{N} K_\kappa(\mathbf{x}_i, \mathbf{x}_j) \right) \tag{2}$$

and the local differential entropy (only between the set $\mathcal{P}(i)$ of positive pairs of $i$) as:

$$H_{\text{local}} = -\frac{1}{N} \sum_{i=1}^{N} \log \left( \frac{1}{N-1} \sum_{j \in \mathcal{P}(i)}^{N} K_\kappa(\mathbf{x}_i, \mathbf{x}_j) \right) \tag{3}$$

Finally, after adding two $\alpha$ and $\beta$ factors (default to 1), we an express the mutual information formula as

$$I(X;Y) \approx \alpha \cdot H_{\text{global}} - \beta \cdot H_{\text{local}} \tag{4}$$

For generating the views of each sample, we opted to a similar augmentation recipe as DINO (Caron et al., 2021). In our case, we used two global views and six local views. Each global view takes a crop of 40% to 100% of the sample. And each local view takes a crop of 5% to 40% of the sample.

Unless mentioned otherwise, all trainings were done using AdamW optimizer, a batch size of 128 and a learning rate of $1e - 3$. To make our runs reproducible, we implemented our code using the Lightly Framework and used the evaluation setup developed by Kalapos & Gyires-Tóth (2024). Models trained on LC25000 were trained for 25000 epochs, the ones trained on ImageNet-1000 were trained for 100 epochs, and the ones trained on CIFAR-10 and STL-10 were trained for 200 epochs.

## 4 MAIN RESULTS

We trained and evaluated our method on CIFAR-10 (Krizhevsky & Hinton, 2009), STL-10 (Coates et al., 2011), LC-25000 (Borkowski et al., 2019) and ImageNet-1000 (Deng et al., 2009) datasets.

| Dataset | Method | Arch | Linear | | K-NN | |
|---------|--------|------|--------|--------|--------|--------|
| | | | Top 1 | Top 5 | Top 1 | Top 5 |
| STL-10 | Supervised | RN18 | 71.86 | 97.41 | 72.43 | 87.69 |
| STL-10 | *Ours* | RN18 | 81.98 | 99.25 | 76.85 | 93.83 |
| STL-10 | *Ours* | VIT-T16 | 85.98 | 99.66 | 80.31 | 94.65 |
| STL-10 | VICReg | RN18 | 81.69 | 99.14 | 78.48 | 93.85 |
| STL-10 | SimCLR | RN18 | **82.89** | **99.36** | **79.16** | **94.91** |
| STL-10 | Barlow Twins | RN18 | 82.19 | 99.04 | 78.20 | 93.98 |
| STL-10 | BYOL | RN18 | 81.28 | 99.28 | 77.14 | 94.39 |
| STL-10 | DINO | RN18 | 79.75 | 99.11 | 76.85 | 93.53 |
| CIFAR-10 | Supervised | RN18 | **84.17** | **98.97** | **84.16** | **93.36** |
| CIFAR-10 | *Ours* | RN18 | 72.54 | 97.97 | 68.11 | 89.14 |
| CIFAR-10 | VICReg | RN18 | 74.72 | 98.14 | 72.67 | 90.59 |
| CIFAR-10 | SimCLR | RN18 | 72.58 | 97.86 | 69.16 | 90.25 |
| CIFAR-10 | Barlow Twins | RN18 | 74.63 | 97.91 | 71.81 | 90.87 |
| CIFAR-10 | BYOL | RN18 | 69.37 | 97.81 | 65.54 | 89.74 |
| CIFAR-10 | DINO | RN18 | 72.46 | 98.02 | 70.07 | 90.34 |
| LC-25000 | Supervised | RN18 | **99.83** | - | **100.00** | - |
| LC-25000 | *Ours* | RN18 | 99.75 | - | 99.99 | - |
| LC-25000 | *Ours* | VIT-T16 | 98.54 | - | 99.89 | - |
| LC-25000 | VICReg | RN18 | 99.30 | - | **100.00** | - |
| LC-25000 | SimCLR | RN18 | 98.35 | - | 99.99 | - |
| LC-25000 | Barlow Twins | RN18 | 98.09 | - | 99.98 | - |
| LC-25000 | BYOL | RN18 | 95.94 | - | 99.54 | - |
| LC-25000 | DINO | RN18 | 98.46 | - | 99.99 | - |
| IM-1000 | *Ours* | RN50 | 52.12 | 77.81 | 35.37 | 64.50 |
| IM-100* | *Ours** | RN50 | 68.15 | 89.68 | 55.25 | 77.38 |

Table 1: Training results. Linear and K-NN accuracies (top-1 and top-5). *IM-100 metrics were calculated using the ImageNet-1000 trained model.

The network architectures trained were ResNet-18 (He et al., 2016), ResNet-50 (for ImageNet-1000) and ViT-Tiny (Dosovitskiy et al., 2021) (for STL-10 and LC-25000). The training was done on a single NVIDIA H100 GPU with 80GB of GPU memory hosted in the cloud. The classification accuracy is summarized in Table 1. While the results were good for datasets with a low number of "latent classes", it is clear that our method did not fare well for more diverse datasets such as ImageNet-1000. However, the gap between top 1 and top 5 accuracy suggests that despite the low top 1 accuracy, our method still produced a good coarse structure. We can confirm this if we take our ImageNet-1000 model and evaluate its linear probe and K-NN accuracy on the ImageNet-100 dataset: the results are much better than in ImageNet-1000, even if the gap is smaller.

### 4.1 EMERGENT FOCUSING ON OBJECTS

Neural networks trained using our method learned to focus on main objects in an image. This behavior was detected both in the ResNet50 trained on ImageNet-1000 and a ViT-Tiny trained on STL-10. As shown in Figure 1, when visualized the gradients using SmoothGrad-CAM technique applied to the ResNet-50 trained on ImageNet-1000, we can see that the network learned with some success to focus on the main objects in each image. On the other hand, in Figure 2, we visualized the average CLS-to-patch attention across all transformers blocks of a ViT-Tiny trained on STL-10 using our method. In both cases, we can see that the networks learned to focus on the main objects in the image.

## 5 IMPACT OF HYPERPARAMETERS

We explored variations of the number of dimensions in the projector, the impact of different $\kappa$ (bandwidth parameter) in the final accuracy and the weight of the local entropy estimation ($\beta$ parameter).

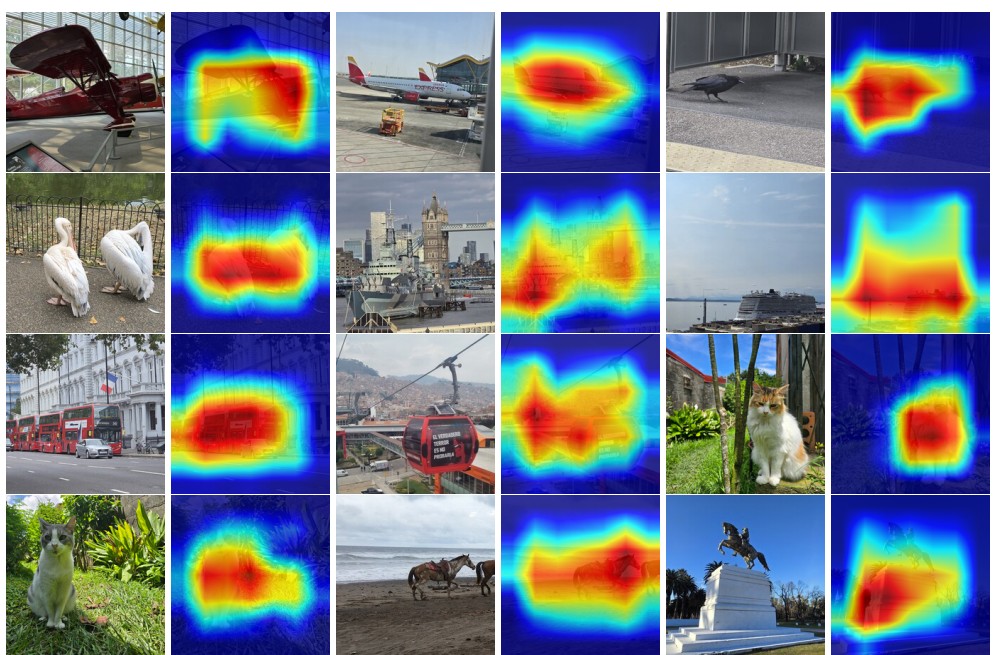

Figure 1: SmoothGrad-CAM visualization of gradients produced by a ResNet-50 trained on ImageNet 1000. Example images are out of training distribution.

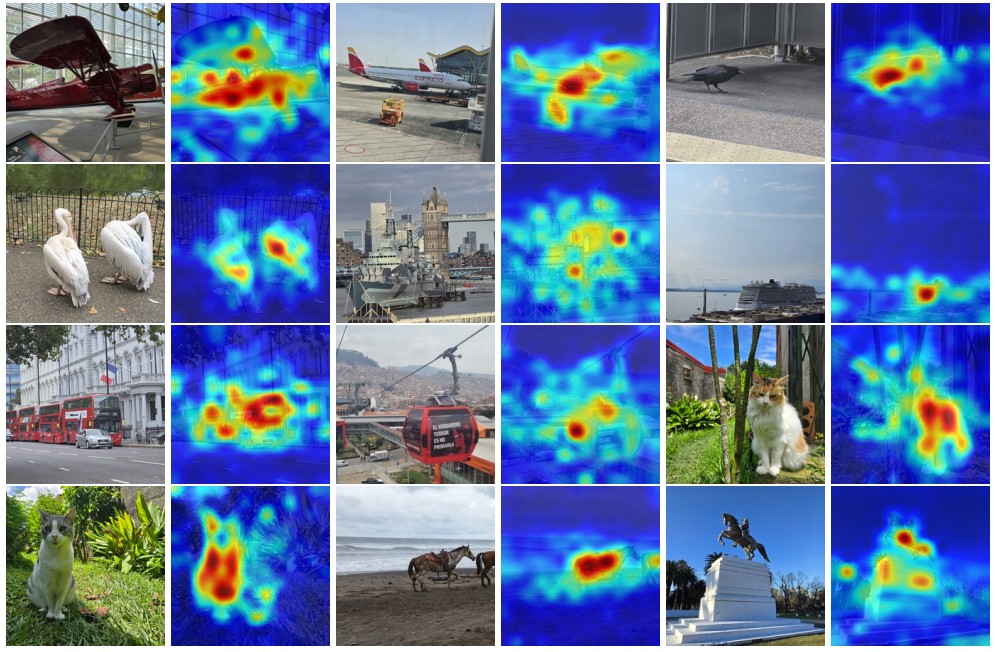

Figure 2: Attention overlays from a ViT-Tiny trained on STL-10, obtained by averaging CLS-to-patch attention across all transformer blocks. Example images are out of distribution.

Unlike other methods such as Barlow Twins, we did not find any noticeable difference for changing the number of dimensions of the projector. On the other hand, $\kappa$ influence is more noticeable.

## 5.1 PROJECTOR DIMENSIONS

We trained a ResNet-18 using our method on STL-10, changing the number of dimensions of the final projector to 128, 256, 512, 1024, 2048 and 4096. There is no clear advantage of increasing the size of the final projector, and the learning curves are very similar in all cases, as shown in Figure 3.

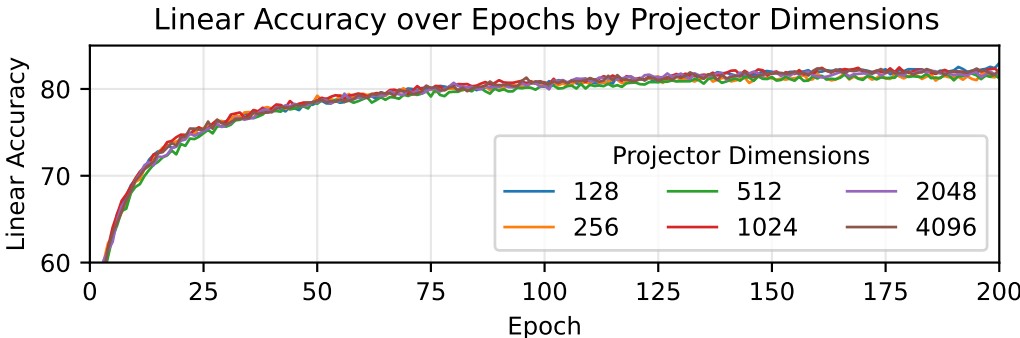

Figure 3: Linear probe top 1 accuracy per epoch, varying the number of dimensions of the projector.

## 5.2 BANDWIDTH PARAMETER

We trained a ResNet-18 using our method on STL-10, varying the bandwidth parameter $\kappa$ to 0.1, $0.6931 \approx \ln 2$, 1, 10 and 20. Recall that $\kappa$ is the concentration (bandwidth) of the vMF kernel used in our non-parametric density estimate: larger $\kappa$ produces a sharper kernel and a smaller $\kappa$ a smoother one. In this case, we found that higher $\kappa$ values accelerate early accuracy, however, if it is too high it hinders the learning process after the initial epochs. Smaller $\kappa$ values lead to slower accuracy improvements, but given enough epochs seems to eventually catch up. In the Figure 4 the accuracy per epoch is shown for each $\kappa$ evaluated.

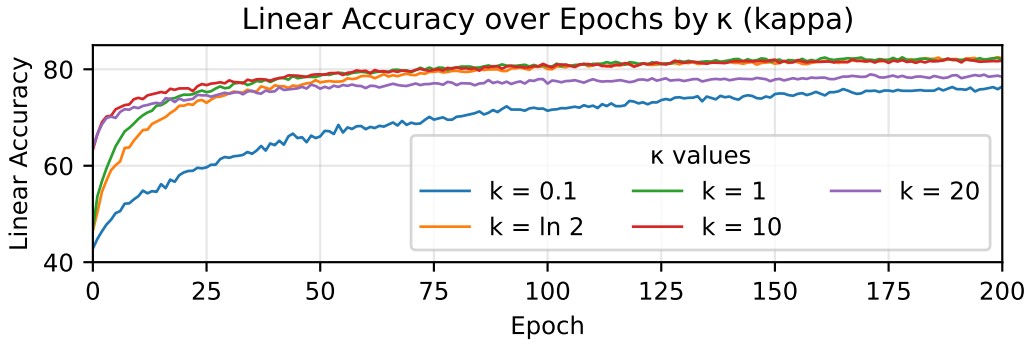

Figure 4: Linear probe top 1 accuracy per epoch, varying the bandwidth parameter.

## 5.3 LOCAL ENTROPY ESTIMATION WEIGHT

Additionally, we modified the weight of the local entropy estimation. We found that reducing the $\beta$ weight, alternatively weighting separation more, leads to higher early accuracy. However, these setups seems to get stuck after some epochs, and runs with higher local weight end up having a higher final accuracy, despite the slower early accuracy gains. The accuracy during training of those variations is shown in Figure 5. Below in that figure, a zoom in section is shown, where we can

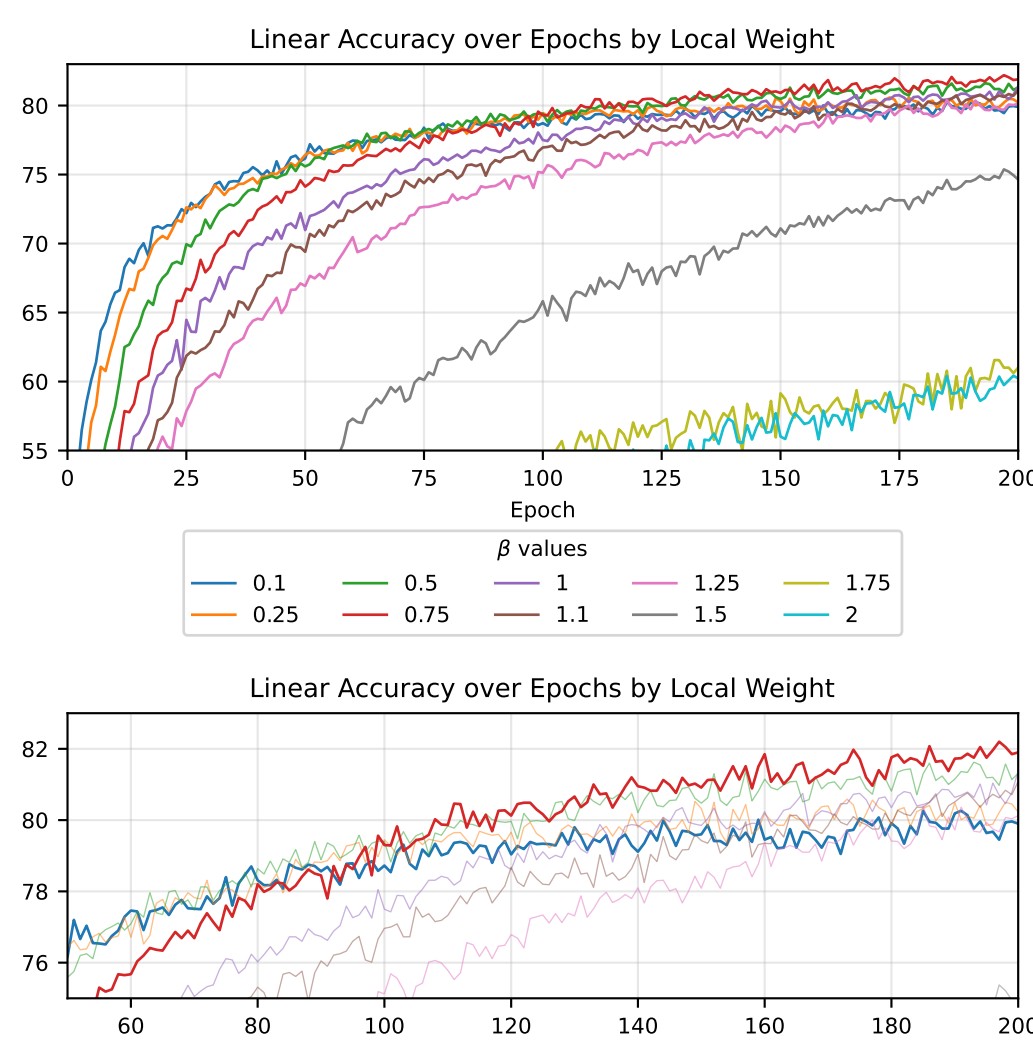

Figure 5: Linear accuracy over epochs by changing the $\beta$ weight (above). Below, a zoom in, highlighting the curves for $\beta = 0.1$ and $\beta = 0.75$

appreciate how the accuracy of $\beta = 0.1$ stalls compared to other settings, particularly $\beta = 0.75$. This suggests that longer training would benefit from a softer separation force. Which is interesting, as this suggests that at least for this sort of classification problems, the best representation may not be the one with maximum mutual information.

## 6  CONCLUSION

In this paper, we discussed an implementation of the InfoMax idea of maximizing Mutual Information. In this case, we opted to use hypersphere geometry, non-parametric kernel density estimators and differential Shannon entropy. We found that this method can generate good representations in datasets with low number of classes, but it starts to lag behind for datasets with higher level of classes. However, the metrics suggest that lower accuracy is due to a learned coarse structure, and not a failure to learn good representations. Our experiments additionally point in the direction that we can get better representations if we relax our global entropy term. This suggests that, at least for classification, representations should not look to maximize mutual information, but instead there

may be another theoretical objective that may explain better how to produce strong fine-grained structures.

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

## A    LEARNING CURVES

The curves for linear top-1 accuracy are shown at Figure 6. It is interesting to note that no method surpassed CIFAR-10 supervised classification, while in STL-10 case, supervised classification got stuck around 70% linear top-1 accuracy.

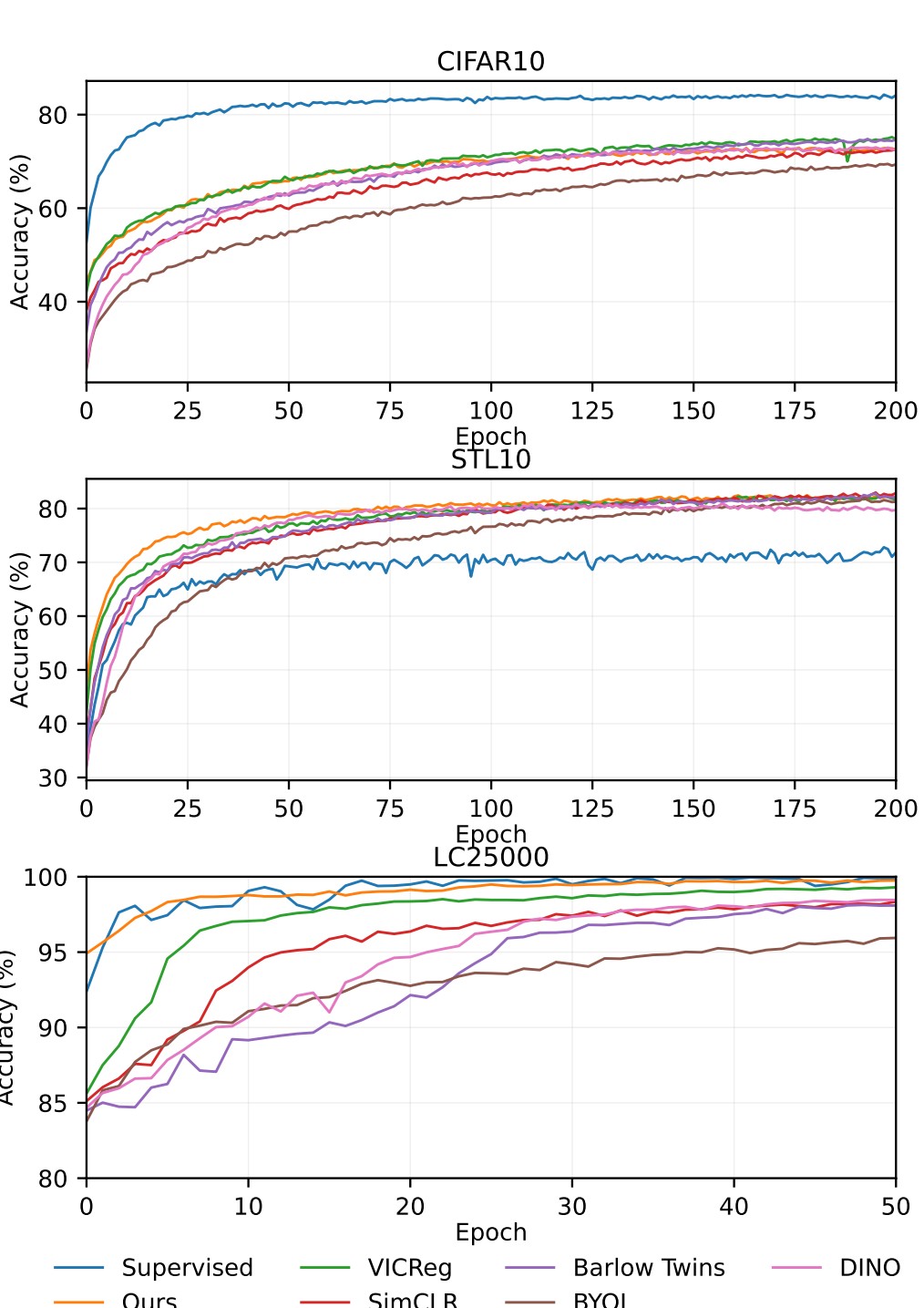

Figure 6: Top 1 accuracy of a linear classifier trained at the end of each epoch.

# B    IMAGENET-100 CLASS CENTROID SIMILARITY

To visualize how the learned representations may overlap between classes, we opted to create a heatmap of the cosine similarity of the class centroids of ImageNet-100 classes, shown in Figure 7. For this, we used a ResNet-50 trained on ImageNet-1000. From this visualization, we can infer that the model has some classes with more overlap than others. For instance, the big square in the middle corresponds to animals. That square itself can be divided in dogs and other animals, being the dogs the most intense square in the lower right corner, and other four legged wild animals above at the left. The rest of the animals are in the weaker big square at the right bottom of the visualization. The weak square at the top left is dominated by human created objects.

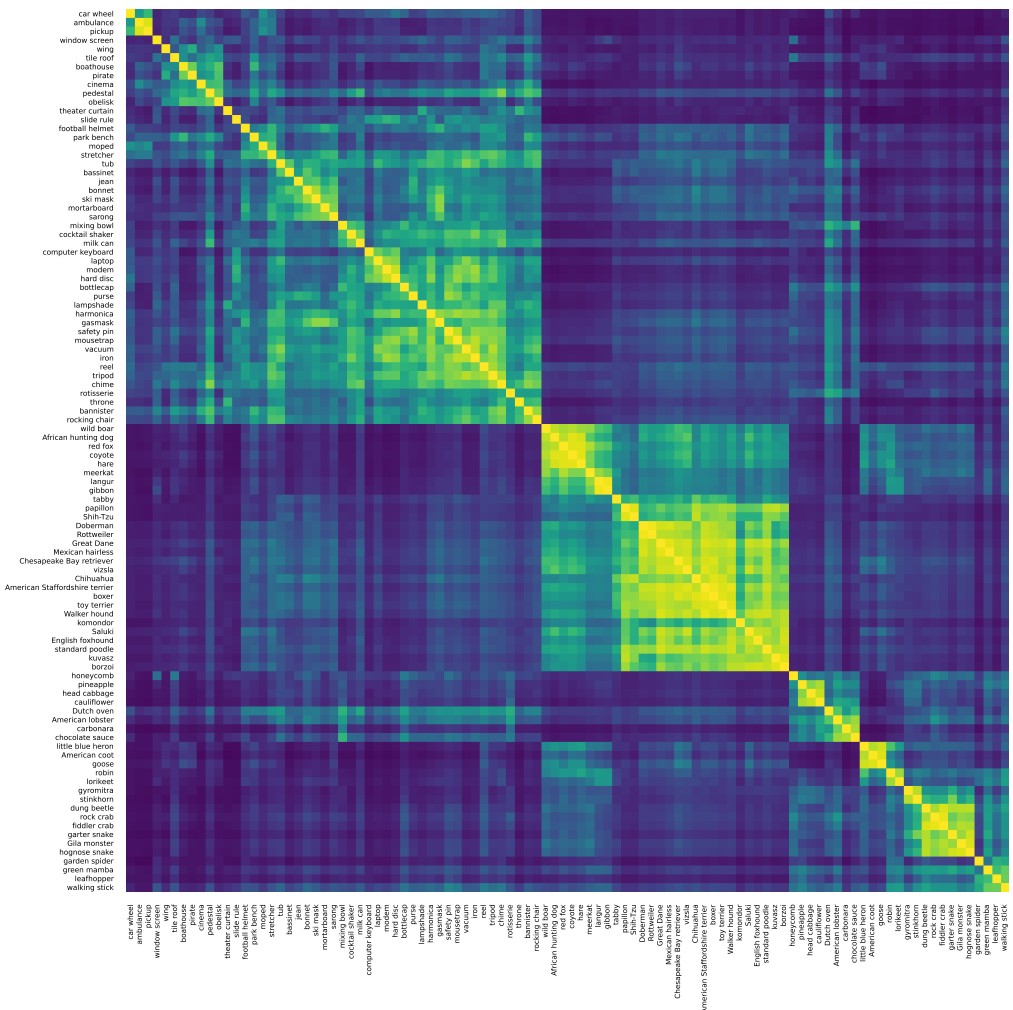

Figure 7: Cosine similarity of ImageNet-100 class centroids, calculated using a ResNet-50 trained on ImageNet-1000.

