# OpenReview forum: "Learning Coarse-Grained Representations: An Exploration of Mutual Information via Hyperspherical Density"
_ICLR.cc/2026/Conference — ICLR 2026 Conference Withdrawn Submission_

### Official Review · Reviewer_AFGX · 2025-10-22

**Soundness:** 3
**Presentation:** 3
**Contribution:** 3
**Rating:** 6
**Confidence:** 3

**Summary:**

The work deals with self-supervised learning of visual representation, exploring the InfoMax principle with a restricted hyperspherical geometry. The densities to compute the entropy are estimated with the  von Mises-Fisher non parametric kernel density estimator and the mutual information approximated by a difference of the global differential entropy and the local one.

**Strengths:**

* while several works seem quite close from the proposed one (see weaknesses) it seems that the use of a direct non-parametric density estimation wit vMF and an approximation of the mutual information by a difference of differential entropies has never been proposed. It is theorically new *and* interesting.

* a large part of the paper is dedicated to the ablation of the approach. Three hyperparameters are studied quantitavelly and qualitative results are also reported.

* the full code is provided as supplementary material, ensuring that it will be released to the community to favour reproducibility. The code is particularly well written and structured.

**Weaknesses:**

* the proposed work should position itself:
  - in relation with [b], where two properties of contrastive representation learning is studied on the Hypersphere.  After their Theorem 1, they relate their work with  feature distribution entropy estimation (via a von Mises-Fisher kernel density estimation) and the InfoMax principle.
  - in relation with [d] which interpret DINO features (the same as those used in the paper) as a von Mises-Fisher mixture model

* the quantitative evaluation are reported with a ResNet-18 backbone only, except of the proposed approach that is estimated with a ViT-16 backbone. One can regret that this last backbone is not used for baseline approaches.

**minor**:
  - line 030: SSL is not defined, it should be first written on line 024: "Self-supervised learning (SSL) methods..."
  - line 045: ICA is not defined (independent component analysis). By the way, ICA was "grounded" by several principles, including contrast maximization by Comon (1991) and Joint Approximate Diagonalization of Eigenmatrices by Cardoso and Souloumiac in 1993.Regarding the Infomax principle, Bell and Sejnowsky indeed relied on it, but Nadal and Parga showed in 1994 that Infomax was equivalent to redundancy reduction principle, opening the path to its usage for ICA.
  - line 053: MI should be defined previously (mutual information on line 046)
  - line 091: "we an" --> "we can"
  - line 101: "Models trained on LC25000 were trained for 25000 epochs" --> "Models were trained on LC25000 for 25000 epochs"
  - implementation details (line 098-103) may be better placed in section 4. In particular the citation of datasets (currently n lines 107-108) should be before the details currently n line 101-103
  - baseline method in Table 1, while known, should be cited in the text
  - line 395: the actual published paper should be cited rather than the arxiv report for ( Kalapos and Gyires-Tóth, 2024) [a]
  - while different, it may worth to position the work w.r.t [c]

[a] A. Kalapos and B. Gyires-Tóth, "Whitening Consistently Improves Self-Supervised Learning," 2024 International Conference on Machine Learning and Applications (ICMLA), Miami, FL, USA, 2024, pp. 448-453,

[b] Wang and Isola (2020) [Understanding Contrastive Representation Learning through Alignment and Uniformity on the Hypersphere](https://proceedings.mlr.press/v119/wang20k/wang20k.pdf), ICML

[c] Li et al (2024) Probabilistic Contrastive Learning with Explicit Concentration on the Hypersphere, arxiv:2405.16460

[d] Govindarajan et al (2023) DINO as a von Mises-Fisher mixture model, ICLR

**Questions:**

* why the top-5 accuracy is ont reported for LC-25000 in Table 1 ?
* why the performance on ImageNet-1k is not reported for the baselines in Table1 ? At least from previous papers...
* which K is used for K-NN in Table 1?
* what is the value of $\kappa$ for the main results of section 4? Are $\alpha$ and $\beta$ set to one (as said line 091)

**Details Of Ethics Concerns:**

Ethic concern is not addressed but due to the positioning of the paper, there is no reason to do so.

---

> ### Author Response · Authors · 2025-11-23
>
> Thanks for your comments and suggestions!
>
> W1: positioning
>
> Wang & Isola: Yes, it is similar. However, notice that the formula used for expansion is different. In Wang & Isola they maximized distance (which ideally would put points at the other side of the hypersphere), while here we just maximize entropy, which would just encourage an orthogonal position.
>
> Govindarajan et al: To be honest, I was not aware of this paper, so my comment is based on a quick reading. At first I would say it is not related, as in our case we are studying InfoMax, they are studying and improving upon DINO. However, both seem to share a similar collapse bias.
>
> W2: Agree. We didn't train those baselines exclusively because of a lack of compute. We used it mostly for ablations and alternative formulations.
>
> Minor:
> Line 101: oh no! For LC25000 the models were trained for ***just 50 epochs***. The paper is wrong.
>
> Questions:
> Q1: LC25k dataset has only 5 classes, so Top 5 is always 100%
> Q2: We didn't train the baselines on ImageNet-1k just because our compute budget was limited. ImageNet-1k takes a lot of time and GPUs are expensive. We considered that it would be better to spend our GPUs on the ablations rather than ImageNet-1k results (which usually are reported in their own papers).
> Q3: For each KNN we used K=5
> Q4: For the main results we used k=a=b=1.

---

### Official Review · Reviewer_ZWUq · 2025-10-29

**Soundness:** 2
**Presentation:** 2
**Contribution:** 1
**Rating:** 2
**Confidence:** 4

**Summary:**

This paper revisits the InfoMax principle for self-supervised representation learning, proposing a simple implementation based on hyperspherical geometry and non-parametric von Mises-Fisher (vMF) kernel density estimation to compute global and local differential entropies.

**Strengths:**

Simplicity and conceptual clarity in method.

**Weaknesses:**

Limited Novelty and Technical Contribution

- The proposed formulation appears to be a relatively direct combination of established components—namely hyperspherical embeddings, von Mises–Fisher (vMF) kernels, and differential entropy estimation.
- The paper does not present a substantial theoretical advancement or novel derivation beyond existing InfoMax- or kernel-density-based self-supervised learning (SSL) frameworks.

Poor Scalability and Weak Empirical Results

- The performance reported in Table 1 is markedly low, contradicting the paper’s claims; even on datasets with a small number of classes, the method underperforms.
- Figures 1 and 2 omit results for several experimental settings, and there is no comparison against standard SSL benchmarks, which undermines the empirical credibility of the work.
- Similarly, Figures 3 and 4 report results only on small or relatively simple datasets, further limiting the strength of the experimental validation.
- The claim that the learned representations are “coarse-grained” is largely descriptive and lacks rigorous quantitative validation.
- No ablation or analysis is provided to substantiate the connection between this phenomenon and mutual-information behavior in a measurable way.

Lack of Theoretical Grounding

- The paper fails to formally justify how the proposed estimator approximates true mutual information.
- From the perspective of mutual-information estimation, the work does not convincingly demonstrate either theoretical rigor or empirical superiority; a comprehensive comparison against existing MI estimators is essential.
- In contrast to prior MI estimators that do not rely on hyperspherical geometric assumptions, the proposed approach is considerably more restrictive.
- The observation that relaxing the global entropy term improves classification lacks analytical explanation and may merely reflect optimization dynamics rather than a genuine information-theoretic insight.

**Questions:**

- In Equation (4), the formulation no longer corresponds to mutual information when both α and β differ from 1. Could you clarify the rationale for setting these parameters to values other than 1, and how this choice affects the interpretation of the objective?
- What is the basis for claiming that the proposed method learns coarser representations? Could you provide a theoretical explanation of why the proposed loss function encourages such coarse-grained structure, and offer empirical evidence demonstrating that this property indeed emerges in the learned embeddings?

---

> ### Author Response · Authors · 2025-11-23
>
> Thanks for your comments and feedback.
>
> W1: Limited novelty
> Please see answer W1 to YLtS.
>
> W2: Poor scalability and weak empirical results
>
> Regarding the lack of some baselines and using simpler datasets: this was due to budget constraints. We preferred to use our compute in the ablations (and other experiments not reported) rather than in training well known baselines ourselves. ImageNet-1k itself was particularly problematic, as it needs many days to fully train and each run is particularly expensive.
>
> In STL-10 the method seems competitive...
>
> However, I have reasons to believe the results should be somewhat extrapolable to datasets with more classes, as long as those classes have enough differences (see answer Q2 below)
>
> W3: Lack of theoretical grounding
> Our goal was to operationalize the InfoMax principle in a way that remains computationally feasible for high-dimensional continuous representations. We found other entropy formulations problematic for our maximization needs. For instance, in unconstrained Euclidean space it may lead to density estimation instability, exploding norms, or intractable entropy computations. By constraining embeddings to the hypersphere and using a vMF kernel, we obtain a well-behaved, normalized representation space and a tractable non-parametric entropy estimator. The intent was not to propose the theoretically optimal MI estimator, but to adopt a geometrically coherent and practically usable approximation that allows us to study how a near-first-principles InfoMax objective behaves in modern SSL settings.
>
> Q1: We noticed that increasing expansion weight made the model learn faster, which is a desirable feature. However, it also worsens the final quality, so there must be a balance. This signals that while InfoMax is a sound principle, our experiments signal that we may not want to absolutely maximize the entropy as much as possible for useful representations.
>
> Q2: That is suggested by appendix B, where the confused classes are mostly around similar objects in ImageNet-100. Also, notice that top5 classification in ImageNet-1000 is far higher than top1: that signals that the model is confused around similarly looking classes, which is different than being completely wrong.
>
> Finally, we find it after submitting the paper, but if we train a linear probe with 100% of the data we can get 100% top 1 accuracy. Sure, obviously not useful for classification, but it is a strong assertion that each class ended up in one linearly separable plane, which is not a trivial result. So, the problem with Top 1 is not lack of structure, but lack of margins.

---

### Official Review · Reviewer_q5N6 · 2025-10-29

**Soundness:** 2
**Presentation:** 1
**Contribution:** 1
**Rating:** 0
**Confidence:** 5

**Summary:**

The paper proposes using a vMF (von Mises–Fisher) kernel as a mutual-information estimator for self-supervised learning.
The authors evaluate the method on four datasets with Grad-CAM, CLS-to-patch attention analyses, and hyperparameter ablation studies.

**Strengths:**

The paper proposes using a vMF kernel as a mutual-information estimator for self-supervised learning.
The authors evaluate the method on four datasets with Grad-CAM, CLS-to-patch attention analyses, and hyperparameter ablation studies.

**Weaknesses:**

1. The paper proposes using a vMF kernel as an MI estimator for self-supervised learning but does not explain how it differs from existing estimators such as InfoNCE or what advantages it offers.

2. The proposed loss is algebraically similar to the InfoNCE loss, differing mainly in the number of positive pairs and its interpretation as a vMF-based entropy estimator, which weakens the novelty of the method.

3. The results in Table 1 do not demonstrate any clear advantage of the proposed vMF-based self-supervised method compared with other baselines.

4. Figure 1 does not convincingly show that the trained model focuses on the main object, as the Grad-CAM activation is broadly distributed over non-object regions.

**Questions:**

Please see weakness

---

> ### Author Response · Authors · 2025-11-23
>
> Thanks for your feedback and comments, they are appreciated.
>
> W1: Please see responses to YLtS, particularly W1 and Q2. Overall, this is not a new method, it was a study on InfoMax principle. InfoMax itself does not tell us how to calculate the entropy, so we choose a formula/geometry that makes it computable.
>
> W2: No, in this case the expansion loss is treating each sample as a negative pair and pushes everyone away. So, it is closer to Wang & Isola uniformity loss, except that they maximized euclidean distance and we maximized entropy on the hypersphere.
>
> W3: Agree. The intention was not showing InfoMax as a better SSL method, but to probe why/where it stalls compared to what we could get from other methods.
>
> W4: We did not claim that the gradients focus exclusively on the "foreground or main object". Instead, we wanted to show that InfoMax dynamics lead to the emergent behavior of the model focusing the gradient/attention on the foreground/main object. It is a bias, not an absolute.
>
> Overall, we do not think some gradient focusing on the background is an issue by itself: for learned representations we may want the representation to represent some detail of the background too.
>
> Also, other methods end up focusing on the background instead while still giving good accuracy (at least in STL-10). So, a bias towards foreground is not a trivial or obvious result by itself.

---

### Official Review · Reviewer_YLtS · 2025-11-01

**Soundness:** 2
**Presentation:** 2
**Contribution:** 3
**Rating:** 2
**Confidence:** 4

**Summary:**

This paper addresses the InfoMax objective function for self-supervised representation learning. The authors estimate mutual information using a von Mises-Fisher (vMF) kernel density estimator under hyperspherical geometry (i.e. embeddings have unit length). They evaluate their method on several datasets (STL, CIFAR, LC-25000, and ImageNet-1K) using standard data augmentations to form positive pairs. The linear probing results for classification show that the method achieves competitive performance on smaller datasets but falls behind on ImageNet-1K.

**Strengths:**

1- the motivation for using vMF kernels is clear, and the objective function is minimal and theoretically grounded.

2- good empirical insight from ablations on projector dimension and band width parameter.

**Weaknesses:**

1- the paper feels incomplete. The direction and main research question are not well defined. It’s unclear whether the goal is to analyze InfoMax behavior or to improve performance.

2- more experiments, analyses, and discussion are needed. For instance, comparisons to alternative entropy estimators or normalization schemes (e.g., stop-grad, whitening) would add clarity.

3- the study of the bandwidth parameter κ closely parallels the temperature parameter in contrastive learning. This connection could be more explicitly analyzed.

**Questions:**

1- can you analyze how your loss function relates to the alignment and uniformity objectives (Wang & Isola, 2020)? Under similar assumptions, what does your objective correspond to?

2- can you elaborate on the motivation for using vMF kernel estimation instead of InfoNCE or other parametric estimators?

3- is the main goal of your work to achieve strong downstream performance, or primarily to analyze the behavior of InfoMax under certain assumptions?

---

> ### Author Response · Authors · 2025-11-23
>
> Thanks for your feedback, it is truly appreciated.
>
> W1: Analyze Infomax behavior.
>
> Given that several papers comment on the empiricism of SSL methods (for instance https://arxiv.org/pdf/2010.00578), we tried to step back and study what is missing in theoretical grounded methods compared to SOTA ones. So, we choose to study Infomax and evaluate it in modern benchmarks such as ImageNet or testing how the attention/gradients focus on the objects. We considered this evaluation useful, because, even if there are other Infomax papers such as https://arxiv.org/pdf/1808.06670, we did not find any paper showing the ImageNet 1k performance or its gradient/attention heatmaps.
>
> W2: More experiments?
> Overall agree, however notice that the scope would be unfeasible. Regarding the chosen entropy estimator: we picked that one because the mix of vMF kernel AND hyperspheric geometry gave us a clean way to calculate the entropy and avoided some potential pitfalls of other ways to do it. For instance: in euclidean space, we may end up with representations going to infinity, clearly something we do not want. Also, see W1: the focus was studying InfoMax, not improving performance.
>
> W3: temperature?
> That is an interesting observation, thanks for pointing it up.
>
> It may not be noticeable from the paper, but most of the effort went into studying why we were getting better representations from actually reducing the weight of the expansion term. But also, that expansion term was the one driving early classification accuracy gains. A lot of ablations went into this area (clearly without positive results).
>
> Q1: Wang & Isola relationship:
> Under my understanding they would be complementary explanations. Maximizing the entropy leads to uniformity on the hypersphere, which Wang and Isola already noticed improves the representations. But there are a lot of ways to organize data in a hypersphere and still having that data uniformly distributed. I'm not sure how the Wang and Isola uniform loss will bias the resulting structure, but in our paper, our formula ideal separation term would lead to orthogonal projections per view, which is the position that maximizes the entropy, but not the maximum possible euclidean distance.
>
> Q3: It was primarily to study the behavior of InfoMax, to inform future methods possibly based on similar principles. Our main findings would be:
> - Infomax itself leads to some degree of emergent focusing on foreground objects
> - The resulting geometry is coarse. We found this after submitting the paper, but actually the information for fine grained separation IS THERE. We just need to use 100% of the data to train the linear probe. Clearly not useful for general classification, but being able to linearly separate perfectly all the classes of ImageNet-100 is an interesting non-trivial property of the learned structure.
> - Infomax by itself does not result in good fine grained separation
> - Reducing the entropy improves slightly the final learning
> - Increasing the entropy leads to faster early learning, but stalls earlier as well.

---

### Note · Authors · 2025-12-02

**Comment:**

We want to thank all the reviewers for their work reading and providing detailed feedback, which is greatly appreciated.

**Withdrawal Confirmation:**

I have read and agree with the venue's withdrawal policy on behalf of myself and my co-authors.